# SerpinB3: A Multifaceted Player in Health and Disease—Review and Future Perspectives

**DOI:** 10.3390/cancers16142579

**Published:** 2024-07-18

**Authors:** Silvia Cagnin, Patrizia Pontisso, Andrea Martini

**Affiliations:** Department of Medicine, University of Padova, 35123 Padova, Italy; silvia.cgn@gmail.com (S.C.); andrea.martini@aopd.veneto.it (A.M.)

**Keywords:** SerpinB3, oncology, metabolism, fibrosis, biomarker, protection from oxidative stress

## Abstract

**Simple Summary:**

This review highlights SerpinB3’s multifaceted roles in liver disease, from fibrosis, carcinogenesis and immune modulation to cell death protection. In different types of cancer its overexpression correlates with tumor aggressiveness; however, in acute oxidative stress conditions, SerpinB3 promotes cell survival. Novel therapeutic strategies targeting SerpinB3 through its upstream regulators are under development, while its therapeutic potential in acute medical conditions has also been proposed.

**Abstract:**

SerpinB3, a member of the serine-protease inhibitor family, has emerged as a crucial player in various physiological and pathological processes. Initially identified as an oncogenic factor in squamous cell carcinomas, SerpinB3’s intricate involvement extends from fibrosis progression and cancer to cell protection in acute oxidative stress conditions. This review explores the multifaceted roles of SerpinB3, focusing on its implications in fibrosis, metabolic syndrome, carcinogenesis and immune system impairment. Furthermore, its involvement in tissue protection from oxidative stress and wound healing underscores its potential as diagnostic and therapeutic tool. Recent studies have described the therapeutic potential of targeting SerpinB3 through its upstream regulators, offering novel strategies for cancer treatment development. Overall, this review underscores the importance of further research to fully elucidate the mechanisms of action of SerpinB3 and to exploit its therapeutic potential across various medical conditions.

## 1. Introduction

Serine-protease inhibitors (serpins) are a large superfamily of proteins mostly acting as inhibitors of the chymotrypsin family, with some of them having other roles such as inhibiting cysteine proteases [1]. The typical structure of a serpin comprises approximately 350–400 amino acid residues and features three beta-sheets (A, B and C) and nine alpha-helices (A through I). The secondary structure elements are arranged into three main regions: the reactive center loop (RCL), the shutter domain and the helical domain [2]. Of note, some serpins do not exhibit inhibitory activity [2]. Serpins can be both extracellular and intracellular proteins, localizing in various cellular compartments and exhibiting different functions depending on their subcellular localization [2]. Human serpins are classified into nine clades (A to I) as recently described in detail by Janciauskiene et al. [2] and summarized in Table 1.

The biological role played by different members of this family is extensive, as antithrombin (AT or SerpinC1) and plasminogen activator inhibitor-1 (PAI-1 or SerpinE1) exhibit antifibrinolytic and anticoagulant activity [3]. AT also operates as a cardioprotective molecule by upregulating AMP-activated protein kinase pathways [4]. Another important serpin is α-1 antitrypsin (SerpinA1), which can inhibit the neutrophil elastase (NE), thus preventing NE-related injury [5]. It is noteworthy that some serpins do not operate as inhibitors but as hormonal transporters and molecular chaperones [6]. Clade B (or ovalbumin-serpins) consists of 13 serpins which lack the N-terminal signal sequence, thus acting mostly as intracellular proteins [7]. The most-studied members of this clade are SerpinB2 (plasminogen activator inhibitor-2 or PAI-2), SerpinB3/4 and SerpinB5 (maspin). SerpinB2 or PAI-2 in normal conditions is highly expressed in keratinocytes, activated monocytes and placenta, playing a role in fetal development, keratinocyte proliferation/differentiation and monocyte differentiation [8]. This serpin is also involved in host defense against virus infection and the metastasis of head, neck, breast and lung cancer [8]. SerpinB5 or maspin is involved in many anti-oncogenic mechanisms such as inhibiting cell invasion and angiogenesis and promoting apoptosis [8].

SerpinB3, previously known as squamous cell carcinoma antigen 1 (SCCA1) is a highly conserved cysteine protease inhibitor that is normally expressed in the basal and parabasal layers of normal squamous epithelium, where it plays an important role in regulating differentiation of the squamous epithelium [7]. It was initially isolated by Kato and Torigoe [9] in the squamous cell carcinoma of the uterine cervix. What at first was thought to be one single protease was later revealed to be two different isoforms, one neutral (SerpinB3) and one acidic (SerpinB4), which share 98% identity at the nucleotide level and 92% identity in their amino acid sequences, the main difference residing in the active site loop [7]. Both SerpinB3 and B4 belong to ovalbumin-serpins and, from a structural point of view, possess nine α-helices, three antiparallel β-sheets and a hydrophobic c-terminal reactive site loop [7] (Figure 1). The physiological role played by SerpinB3 and B4 remains largely unknown, in part due to the lack of adequate animal models, as there is no genetic match for human serpins in rodents [10]. They are frequently co-expressed in different tissues, including the lung, trachea, prostate, uterine cervix and testis, while in the bladder and thymus only SerpinB3 is expressed. In other tissues, both SerpinB3 and B4 are usually undetectable, but their expression increases in conditions of chronic inflammation, likely as a protective mechanism against cellular stress conditions [7]. Despite these two serpins exhibiting high sequence homology, they are characterized by different biochemical properties and substrate affinities [11]. In vitro, human SerpinB3 mostly targets papain-like cysteine proteases (such as cathepsin L, S and K and papain), whereas SerpinB4 inhibits chymotrypsin-like serine proteases (such as chymase and cathepsin G) [7]. Both SerpinB3 and B4 inhibit proteolytic activity by forming an SDS-resistant complex with the target protease through an acyl-oxyester bond.

Focusing on the topic of this review, SerpinB3 has a protective role in acute damage [12], but its chronic overexpression acts as an oncogenic factor, leading to apoptosis resistance [13,14], cell proliferation and fibrosis (Figure 1). SerpinB3 expression increases in response to Tumor Necrosis Factor (TNF)-α and Ras-driven inflammation [15,16] leading to NF-kB activation, IL-6 production and tumor growth [16,17]. The prevention of apoptosis is achieved through interactions with lysosomal proteases upon lysosomal leakage [10]. Moreover, one of the main properties of SerpinB3 is its ability to induce an epithelial–mesenchymal transition (EMT), facilitating cell invasion and metastasis formation [18]. Recently, Chen et al. [19] highlighted a potential role of SerpinB3 in the regulation of the immune response, favoring an immunosuppressive tumor microenvironment in cervical cancer. 

This review is mainly addressed to analyze the role of SerpinB3 as both a positive and negative molecule, focusing especially on liver disease.

## 2. SerpinB3 in Fibrosis and Carcinogenesis

The first study that connected SerpinB3 and the progression of a fibrogenic chronic disease was made by Calabrese et al. [20] as they investigated the role of SerpinB3 in metaplastic epithelial cells in idiopathic pulmonary fibrosis (IPF), a progressive chronic disease with a poor prognosis. In this study, SerpinB3 was identified as significantly overexpressed in patients with IPF vs. controls, with a positive correlation shown between SerpinB3 levels and the expression of both Transforming Growth Factor (TGF)-β1 and the extension of fibroblastic foci, also suggesting for the first time the possible induction of proliferation and activation of lung fibroblasts in a paracrine way [20].

SerpinB3 has also been related to liver inflammation and fibrosis. Chronic liver disease (CLD) is a major contributor to global mortality, morbidity and healthcare resource utilization [21]. Liver cirrhosis currently results in 1.16 million deaths annually, with liver cancer responsible for 788,000 of these deaths [22,23]. These conditions are among the top twenty causes of death worldwide, accounting for 3.5% of all global mortality [22,23]. Primary liver cancer is the seventh most common cancer and the second leading cause of cancer-related deaths, with hepatocellular carcinoma (HCC) making up over 75% of liver tumors [24,25]. The incidence of HCC increases with age, peaking at 70 years, and shows a 2–3 times higher incidence and mortality in males [24,26,27]. Notably, while mortality rates for other cancers are declining, HCC remains one of the fastest-growing causes of cancer-related deaths worldwide.

In chronic liver disease, SerpinB3 was proven to upregulate the expression of TGF-β1 by directly activating the expression of pro-fibrogenic genes (such as collagen type 1A1, α-smooth muscle actin [α-SMA], TGF-β1, tissue inhibitor of metalloproteases type 1 [TIMP-1], the platelet-derived growth factor B [PDGF-B] and its β receptor [PDGFRβ]) in human liver myofibroblasts in vitro [28,29]. In the same study by Novo et al. [29], SerpinB3 promoted the oriented migration of the myofibroblast-like cells in a reactive oxygen species (ROS)-dependent manner through the activation of Akt and c-Jun-aminoterminal kinases (JNK). TGF-β is also involved in impaired immune response, and its upregulation by SerpinB3 requires the integrity of the antiprotease activity, as deletions in the reactive site loop of this serpin inhibit this effect [28], whereas a single amino acid substitution (Gly351Ala) in the reactive center loop of the protein, such as in the polymorphic variant SCCA-PD or SerpinB3-PD (SB3-PD), determines a gain of function [30]. The gain of function attributed to SB3-PD was observed to be particularly potent in inducing higher expression of TGF-β in HepG2 and Huh-7 cells, leading to increased levels of both inflammatory and fibrogenic cytokines [31]. Furthermore, SB3-PD exhibited greater efficacy compared to its wild-type counterpart also as a paracrine mediator, inducing higher levels of TGF-β in both human stellate cells and THP-1 macrophages. Notably, in THP-1 cells, SB3-PD induced more prominently a mixed M1/M2 profile [31]. In addition, a cohort study involving outpatients with advanced chronic liver disease was also carried in the same report, documenting that patients carrying the SB3-PD variant had signs of a more severe portal hypertension and a higher incidence of both first episodes of decompensation and then further episodes of cirrhosis complications [31].

Of note, the expression of TGF-β and SerpinB3 has been related to the activation of the WNT/β-catenin pathway in both hepatocellular carcinoma (HCC) and colorectal cancers, associating with more aggressive tumors, with an earlier recurrence and a worse prognosis [32,33]. The importance of the WNT/β-catenin pathway lies in its role in embryogenesis, cell renewal and tissue homeostasis but also in tumor growth and dissemination [34,35]. SerpinB3 has been linked with the WNT pathway as it induces the overexpression of β-catenin and the Myc oncogene, a downstream gene of the WNT pathway [36]. The WNT pathway can also be upregulated by SerpinB3 through the overexpression of the low-density lipoprotein receptor-related protein (LRP) family, in particular LRP-1, LRP-5 and LRP-6 whose upregulation leads to an increased β-catenin translocation in the nucleus [37]. LRPs, especially LRP-5 and LRP-6, are crucial co-receptors for the activation of the canonical WNT-signaling. When phosphorylated, axin is recruited to the cytoplasmic tail of LRP-6 and prevents β-catenin phosphorylation and proteasomal degradation, leading to its accumulation in the cytoplasm and subsequent translocation to the nucleus [38]. SerpinB3 is also able to upregulate LRP-1, which is involved in carcinogenesis through its promotion of cell migration, invasion and survival [39].

Interestingly, as hypoxic conditions have been linked to the progression of fibrosis and chronic liver disease [40,41,42,43,44,45], a possible correlation with SerpinB3 levels was investigated. Hypoxia Inducible Factor (HIF)-1α and -2α are the main players in cell response to hypoxia, with HIF-1α being involved in cell proliferation, metabolic changes, angiogenesis and metastasis [46,47,48,49,50,51] and HIF-2α being involved in cell proliferation, resistance to radio- and chemotherapy, self-renewal capability and stem cell phenotype in non-stem-cell populations [52,53,54,55,56,57]. Hypoxic environments stimulate a higher expression of SerpinB3, with HIF-2α directly binding to its promoter [58]. Foglia et al. [59] found a strong association between HIF-2α and SerpinB3 in human specimens of HCC, with HIF-2α being positively related with an increased YAP and c-Myc signaling. Furthermore, SerpinB3 was able to inhibit c-Myc degradation and to increase YAP expression leading to an activation of the Hippo pathway [60], also acting as a paracrine mediator by upregulating even in normoxic conditions both HIF-1α and HIF-2α [61]. Thus, the transcriptional upregulation of HIF-1α supports cell survival in hypoxic environments by inducing an early cellular metabolic switch to the glycolytic phenotype, and the stabilization through NEDDylation of HIF-2α has been proposed as a mechanism to promote cell proliferation in liver cancer [61].

The pro-fibrogenic role of SerpinB3 was also investigated both in vitro and in animal models of metabolic dysfunction-associated steatotic liver disease (MASLD) and metabolic dysfunction-associated steatohepatitis (MASH). In a study by Novo et al. [62], transgenic mice either overexpressing SerpinB3 or carrying a deletion in the reactive site loop were fed a methionine- and choline-deficient (MCD) diet or a choline-deficient and amino-acid-refined (CDDA) diet to induce MASLD. In these experiments, mice overexpressing SerpinB3 showed a marked increase in macrophage infiltrates and a higher level of pro-inflammatory cytokines, whereas these changes were not evident in knockout mice [62]. Additional in vitro experiments exposed phorbol-myristate acetate-differentiated human THP-1 macrophages to SerpinB3, leading to an increased production of M1-cytokines (TNF-α, IL-1β, TGF-β1, vascular endothelial growth factor [VEGF] and ROS) through the activation of NF-kB. In a murine model of MASH, genetically modified mice with a SerpinB3 defective in the reactive site loop showed less TGF-β expression and a reduced macrophage infiltration in the liver [62]. Moreover, transgenic mice overexpressing SerpinB3 presented a higher expression of TGF-β, an increase in Triggering Receptor Expressed on Myeloid cells (TREM)-2 infiltration [62] which is also associated with the severity of steatosis, inflammation, hepatocyte ballooning and fibrosis [63]. In relation to patients with MASLD and MASH, SerpinB3 is also able to deeply affect lipid metabolism, as the stabilization of HIF-2α plays a role in the regulation of hepatocellular lipid accumulation [59,64,65,66]. The interplay between SerpinB3 and HIF-2α has been previously discussed. A study by Foglia et al. [59] highlighted that a specific deletion of HIF-2α in a rodent model of MASH-related liver carcinogenesis led to a significant reduction in volume and number of liver tumors vs. controls. In this experiment, there was a reduction at the nuclear level of Ki67, a marker of cell proliferation, and a downregulation of both Myc and YAP expression [59]. The close relation between SerpinB3, the levels of HIF-2α and their role in the modulation of the YAP/Myc pathway during carcinogenesis in MASH patients highlights the potential of SerpinB3 as novel therapeutic target. These considerations could also be expanded to MASLD and MASH, where the involvement of SerpinB3 is relevant, and MASLD is emerging as one of the major causes of chronic liver disease, especially in patients with obesity and type II diabetes [67,68].

As mentioned before, the role of SERPINs in carcinogenesis was initially studied in squamous cell carcinoma (SCC) of the uterine cervix, in which SerpinB3 and B4 were first identified [9,69]. Later, SerpinB3 was found to be highly expressed in various types of squamous cancers other than that of the uterine cervix, including head and neck, breast, esophageal, and primary liver cancers (HCC, cholangiocarcinoma [CCA] and hepatoblastoma [HB]), being associated with poor prognosis and a higher risk of recurrence [70,71,72,73,74,75]. Of note, SerpinB3 is physiologically expressed in the lung, the esophagus and the uterine cervix, whereas it is almost undetectable in normal hepatocytes but was found to be overexpressed in HCC [58,76] as well as in highly dysplastic nodules and in hepatocytes surrounding the tumor, suggesting that its overexpression represents an early event in liver carcinogenesis [28,58,76,77,78].

The role of SerpinB3 in carcinogenesis includes the induction of EMT [18,79,80], the ability to inhibit cell death by preventing cancer cell apoptosis [81] by either inhibiting JNK or P38 mitogen-activated protein kinase (MAPK) and/or suppressing mitochondrial ROS generation (Figure 2) [14,82,83]. In particular, the localization at the inner mitochondrial compartment allows SerpinB3 to bind to the respiratory Complex I and inhibit ROS generation, preventing or reducing the opening of the mitochondrial permeability transition pore (MPTP), thus protecting cells from the toxicity of pro-oxidant chemotherapeutic agents such as doxorubicin and cisplatin [14]. As aforementioned, the inhibition of lysosomal proteases was proposed as an additional carcinogenic mechanism, through the induction of a constitutive and chronic activation of the endoplasmic reticulum stress-related unfolded protein response [81].

Regarding its role in primary liver cancers, SerpinB3 was found to be expressed in aggressive forms of all these tumors, since its expression is observed in the hepatic stem cell compartment of both fetal and adult cirrhotic livers [84].

Turato et al. [85] identified SerpinB3 as a potential target gene for miR-122, the most expressed miRNA in the liver and whose role is crucial for normal liver function [86,87]. miR-122 is downregulated in pre-neoplastic nodules and in HCC and is inversely associated with metastasis formation and poor prognosis, although the underlying mechanisms are still unclear [85]. miR-122 overexpression was associated with lower levels of SerpinB3 due to decreased gene activity, and on the other hand, high levels of SerpinB3 induced a downregulation of miR-122 in both in vivo and in vitro experiments [85]. Regarding its importance in therapy, miR-122 overexpression determined sensitization to Sorafenib in different cell lines, whereas the presence of SerpinB3 overexpression determined resistance to the drug [85].

Especially in highly proliferative and poorly differentiated forms of hepatoblastoma (HB), the most common liver malignancy in childhood, studies reported frequent activations in the β-catenin gene, leading to elevated levels of β-catenin, Myc and cyclin D1 [88,89]. Turato et al. [36] highlighted that SerpinB3 is detectable in most HB cases, with the highest levels being detected in the most aggressive subtypes [90]. As mentioned before, in HB, the upregulation of SerpinB3 was significantly correlated with Myc expression, an effect independent of the presence of the serpin reactive loop [36], potentially due to an interaction with the surface receptor LRP-1 downstream of the reactive site loop [37].

In cholangiocarcinoma (CCA), the second most common primary liver tumor after HCC, cancer stem cells (CSCs) have been identified as a driving force for initiation, dissemination and drug resistance [91,92]. A study by Correnti et al. [93] identified SerpinB3 as a crucial modulator of the stemness features of CCA. Experiments on cultured cells showed that SerpinB3 expression was markedly upregulated in the subset of stem-like cells of CCA that formed 3D spheres, with this subset of cells being able to activate macrophages towards a tumor-associated macrophage (TAM) phenotype, thus inducing a higher tumorigenic potential and stemness features [94,95]. These stemness features were associated with an upregulation of the gene expression of stem-like markers (such as c-Myc, STAT3 and YAP) and ECM remodeling-related genes (such as various isoforms of matrix metalloproteinases (MMP), integrin beta-3, a-disintegrin and metalloproteinase) [93]. These alterations in gene expression led to the activation of key molecular pathways, such as mitogen-activated protein kinases like Extracellular Regulated Kinases (ERK) 1 and 2, p38, JNK-1, the phosphorylation of the p65 subunit of NFkB transcription factor and the upregulation of c-Myc, NOTCH, MMP9 and β-catenin [93]. Moreover, these results were validated in vivo using immune-deficient mice in which CCA cells transfected to overexpress SerpinB3 caused increased tumor formation with higher weight and volume of neoplastic masses when compared with controls [93]. In human intrahepatic CCA, the presence of high levels of SerpinB3 was associated with lower survival and a shorter time to recurrence [93,96,97], and these findings are in line with preliminary results described in specimens of extrahepatic CCA in which the presence of high levels of SerpinB3 in the bile compartment was associated with a higher frequency of portal invasion and a higher rate of tumor recurrence after surgery [98].

Regarding its role in the immune response, in cervical tumors, SerpinB3 was found to protect neoplastic cervical cells against radiotherapy (RT)-induced damage by preventing lysoptosis [99], and patients with persistently high levels of SerpinB3 before and during RT had a higher risk of recurrence and death [73]. Cervical cancers with higher levels of SerpinB3 secrete higher levels of chemokines that attract myeloid cells, which have an immunosuppressive activity through inhibition of T-cell activation, thus interfering with RT-induced antitumor immunity [19]. The high expression of SerpinB3 in these neoplastic cells was also associated with an increase in phosphorylated STAT3, further leading to an immunosuppressive environment through cell-intrinsic and -extrinsic mechanisms as in other cancer types (head and neck, lung) [100,101,102]. Higher expressions of STAT3 inhibit immunogenic chemokine production, induce the expression of PD-1/PD-L1 and regulate suppressive immune activities in immune cells [103,104,105]. Reduced tumor sensitivity to chemotherapy and an impairment in the immune surveillance induced by high SerpinB3 expression was also demonstrated in esophageal carcinoma with a poor prognosis [74]. In glioblastoma, SerpinB3 was found to drive cancer stem cell survival, whereas in breast and ovarian cancer, it promotes oncogenesis and resistance to chemotherapy [72,106]. Ohara et al. [107] also demonstrated that the SerpinB3-Myc axis is upregulated in the basal-like/squamous subtype of pancreatic cancer. In melanoma, SerpinB3 was the most significant response-related gene for immune checkpoint blockade therapies [108].

## 3. SerpinB3 as a Promising Protective Molecule

With its role in cell proliferation, EMT and cell death regulation, SerpinB3 was also studied for its potential role as a protective molecule, especially in acute stress conditions, as described in Figure 3.

In diabetic ulcers, SerpinB3 has been recently found to be involved in successful healing due to its role in fibrogenesis and angiogenesis [29,61]. SerpinB3 was found to be markedly downregulated in non-healing diabetic wounds when compared to rapidly healing wounds [109,110], and a study by Albiero et al. demonstrated that the local administration of SerpinB3 through a wet silica gel was successful in delivering the protein to the outer skin layer and in improving ulcer healing [111].

The involvement of SerpinB3 in immune modulation was also studied in a murine model of systemic lupus erythematosus, in which the administration of SerpinB3 resulted in increased levels of Tregs in the spleen, leading to a more tolerant immune phenotype and slower disease progression [112].

In the liver, as SerpinB3 levels increase in hypoxic environments, the potential role of this protease was studied in ischemia/reperfusion (I/R) injury. I/R injury occurs after liver resection, transplantation or hemorrhagic shock, with hypoxia and reoxygenation being two essential phases of the process. An increase in SerpinB3 levels was also found to be a positive biomarker after hepatic resection, as hypoxia and oxidative stress can induce the release of SerpinB3, thus conferring resistance to apoptosis, reducing oxidative stress and adding a stimulus for liver cell proliferation [14,113,114]. This biological effect is likely achieved through direct interaction of SerpinB3 with the intramitochondrial respiratory complex I, leading to a reduction in ROS generation [14]. Moreover, as mentioned before in this review, the induction of SerpinB3 by HIF-1α and HIF-2α further supports cell survival in hypoxic environments. HIF-2α also protects against acute liver injury through the production of IL-6 [115].

The increased levels of IL-6, an acute phase reactant cytokine, found in this environment are also associated with high levels of SerpinB3 through its direct binding to the promoter [116,117]. IL-6 indeed activates STAT3 that in turn activates genes that induce liver regeneration [116,118,119,120], and the binding to the promoter of SerpinB3 leads to the induction of a positive loop [121].

## 4. The Future: A Novel Druggable Target for SerpinB3 Inhibition?

A member of the protease-activated receptors (PARs) family, namely PAR2, has been linked to stress responses such as cell proliferation, differentiation and EMT in gastrointestinal and pancreatic cancers [122,123]. PAR2 is also involved in cholesterol homeostasis and lipid metabolism and in suppression of glucose internalization, glycogen storage and insulin signaling [124,125]. PAR2 is activated by trypsin-like proteases, such as tryptase released by mast cells [126,127], matriptase [128] and coagulation factors VIIa and Xa [128,129], when there is an upregulation of tissue factor expression [130] such as in subjects with fibrotic liver disease. The interaction of PAR2 with these proteases establishes a microenvironment capable of initiating prolonged activation of the PAR2 signaling pathway, which encompasses the stimulation of MAPK associated with inflammation, proliferation and mesenchymal cell differentiation via pathways involving IL-1β, TNF-α, TGF-β and NFκB [131].

Targeting PAR2 has been a challenge, due to the continuous changing of its conformation in response to inflammatory proteases. Different antagonistic compounds have been developed, such as peptides, peptidomimetics, cell-penetrable pepducins, small molecules and antibodies [132]. While the majority of them are at preclinical stage, only a small minority of these new drugs has entered the clinical phase [132]. Many problems are currently faced in the development of molecules targeting PAR2, such as protease promiscuity, since proteases bind with different affinities and offer unique PAR2 cleavage, making pharmacological targeting problematic. In addition, heterodimerization between PAR1 and PAR2 forces the development of hetero-bivalent ligands to inhibit signaling activation [132].

In a recent study, the small molecule 1-piperidine propionic acid (1-PPA, PubChem CID 117782) was able to inhibit PAR2, thus blocking a positive loop that involves the upregulation of the early transcription factor CCAAT Enhancer Binding Protein beta (C/EBP-β) and the subsequent promotion of SerpinB3 transcription [133]. In this study, C/EBP-β has been indeed reported as one of SerpinB3 transcription factors and plays a role in metabolic syndrome. In particular, mice lacking the active form of SerpinB3 had at basal conditions not only lower levels of C/EBP-β and a decreased fat mass but also presented a lower inflammatory response after an MCD or CDAA diet, whereas transgenic mice overexpressing SerpinB3 presented higher levels of C/EBP-β than those of control mice, highlighting the essential role of the anti-protease activity of this serpin to achieve this effect. Notably, 1-PPA did not induce significant cell and organ toxicity, while inhibiting PAR2, C/EBP-β and SerpinB3 synthesis in a dose-dependent manner at very low concentrations [133]. Notably, the precise mechanism of action of this compound has been identified, since it acts as an allosteric inhibitor of PAR2, showing the ability to stabilize the receptor in an inactive conformation, even at high temperatures and therefore blocking the positive loop between SerpinB3-induced PAR2 and C/EBP-β [133].

## 5. Conclusions

SerpinB3 is a serine-protease inhibitor deeply involved in tissue homeostasis both in physiological and pathological conditions, from tissue repair and immune modulation to carcinogenesis and metabolic disorders. While its involvement in fibrosis, carcinogenesis and inflammation underscores its significance as a potential prognostic marker and therapeutic target in liver disease and cancer, its emerging roles in wound healing and tissue repair suggest broader implications in diverse medical conditions. The complex interplay between SerpinB3 and various signaling pathways highlights its relevant regulatory role and underscores the need for further research to better understand its mechanism of action and to unravel its full therapeutic potential. Moreover, SerpinB3 could be used as a potential biomarker for both diagnostic and prognostic purposes, especially in liver disease and cancer. Additionally, the promising findings regarding the therapeutic potential of targeting SerpinB3 through its upstream regulators, such as PAR2, suggest new opportunities for the development of novel treatment modalities in cancer and in PAR2-induced diseases.

## Figures and Tables

**Figure 1 cancers-16-02579-f001:**
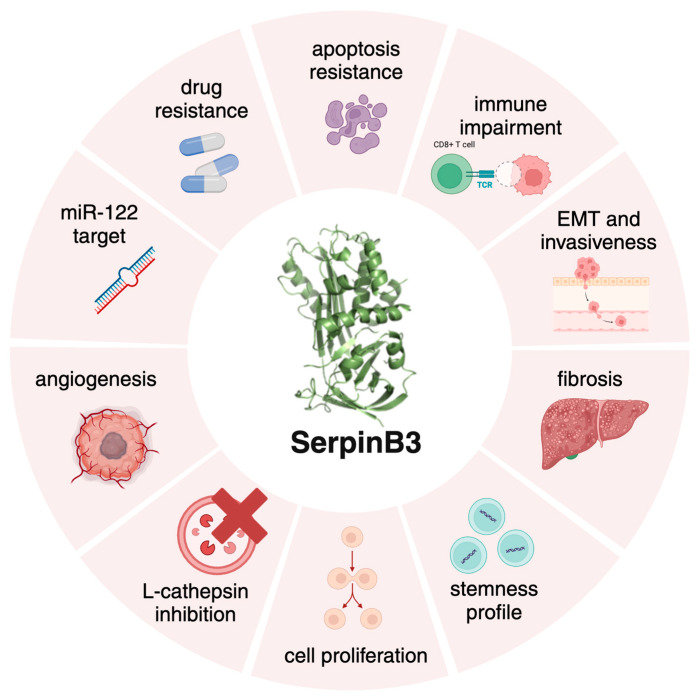
The multifaceted roles of SerpinB3. Graphical representation of the different biological activities of SerpinB3, mainly resulting in cell death protection, fibrosis, carcinogenesis and immune modulation. Created with BioRender.com.

**Figure 2 cancers-16-02579-f002:**
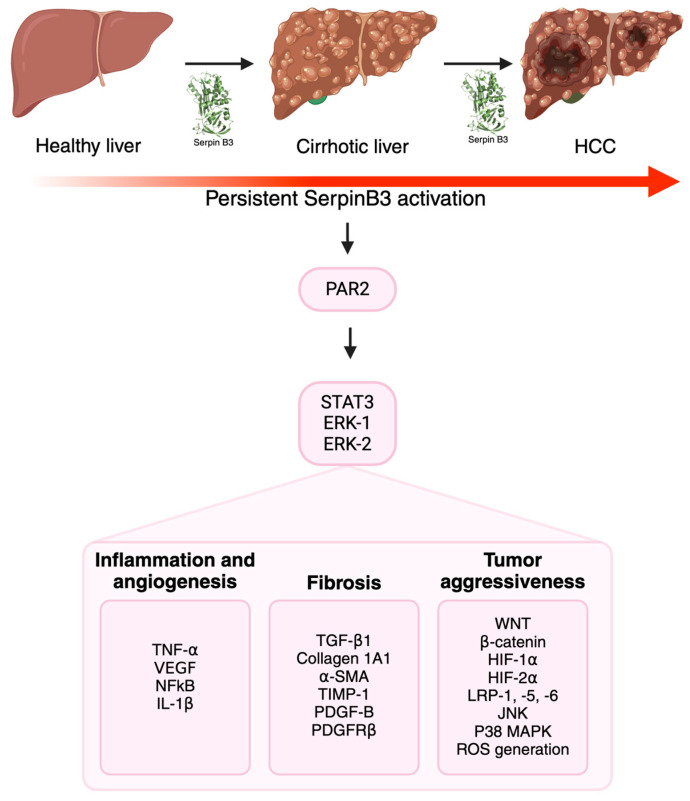
The multiple roles of SerpinB3 in carcinogenesis. SerpinB3 can contribute to the progression of chronic liver disease and cancer development via the release of inflammatory, angiogenetic and pro-fibrogenic mediators. The activation of several intracellular pathways unravels the particular aggressiveness of tumors that express this serpin. Created with BioRender.com.

**Figure 3 cancers-16-02579-f003:**
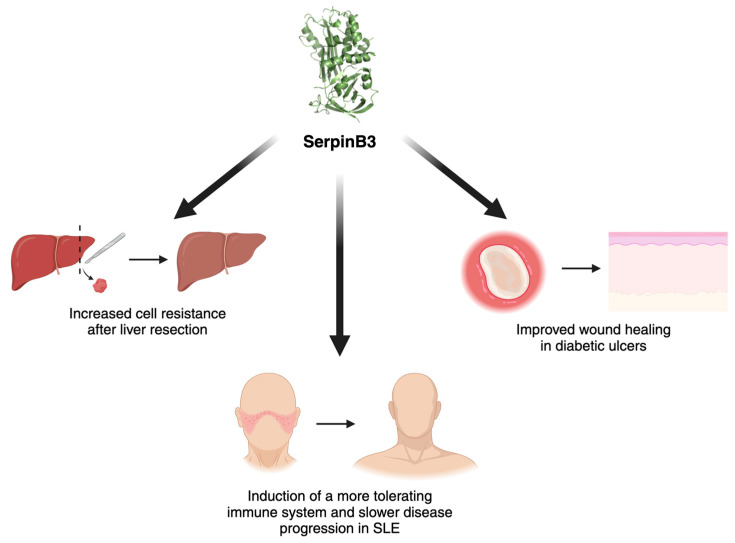
Protective roles of SerpinB3. Graphical representation of the positive effect of SerpinB3 in acute stress conditions and in autoimmune settings. SLE, systemic lupus erythematosus. Created with BioRender.com.

**Table 1 cancers-16-02579-t001:** The nine clades of human serpins. The names, localization and protein names are listed.

Clade Name	Localization	Protein Name
A	Extracellular	α_1_ antitrypsin, antitrypsin-related protein, α_1_ anti-chymotrypsin, kallistatin, protein C inhibitor, centerin, protein Z-dependent proteinase inhibitor, SerpinA11 antiprotease-like, vaspin, SerpinA13, corticosteroid-binding globulin, thyroxine-binding globulin, angiotensinogen
B	Intracellular	Leukocyte elastase inhibitor, plasminogen activator inhibitor-2, squamous cell carcinoma antigen-1, squamous cell carcinoma antigen-2, S proteinase inhibitor, megsin, cytoplasmic antiproteinase 8, cytoplasmic antiproteinase 9, bomapin, epipin, yukopin, headpin, maspin
C	Extracellular	Antithrombin
D	Extracellular	Heparin cofactor II
E	Extracellular	Plasminogen activator inhibitor-1, protease nexin I, serpin family E member 3
F	Extracellular	Alpha-2 antiplasmin, pigment epithelium-derived factor
G	Extracellular	C1 esterase inhibitor, C1 inhibitor
H	Intracellular	Heparin cofactor II
I	Extracellular	Neuroserpin, myoepithelium-derived serine proteinase inhibitor (PI14) pancipin

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
