# Peer review of "SerpinB3: A Multifaceted Player in Health and Disease—Review and Future Perspectives"

_cancers, 2024, doi:10.3390/cancers16142579_

Round 1

Reviewer 1 Report

Comments and Suggestions for Authors

This study focuses on the role of SerpinB3 in liver diseases. While the content is overall significant, the following areas need enhancement before the manuscript is published:

1.     Introduction Section: In the introduction of SERPINs, it should be mentioned that human SERPINs are classified into 37 types across 9 clades. It is necessary to distinguish and summarize the characteristics of each clade.

2.     Clade B Group Characteristics: An introductory text is needed that explains the characteristics of the clade B group in more detail. The similarities and differences between SerpinB3 and SerpinB4 should be explained both structurally and functionally. As they are structurally 98% identical, a functional comparison of SerpinB3 and SerpinB4 is needed, ranging from liver inflammation to liver cancer. It is especially confusing because the content covered in the introduction is somewhat mixed with the content on page 4 (lines 163-165, 165-173).

3.     Focus on SerpinB3: First, summarize its generally known properties and discuss its localization and structural characteristics. Then, describe its role specifically in liver disease.

4.     Introduction to Liver Disease: A brief introduction to liver disease is necessary. Describe the expression pattern and functional significance of SerpinB3 in various diseases and developmental stages. Additionally, the pathways associated with each disease should be presented in a more readable manner.

5.     Gene and Pathway Interactions: The interactions between SerpinB3 and many genes involved in the progression from a healthy liver to liver cancer should be explained by adding Figure 2 or Figure 3.

6.     References: There are too many references. The list should be improved to include more recent studies and reviews. Example: Diagnostic and therapeutic value of human serpin family proteins. Janciauskiene S, Lechowicz U, Pelc M, Olejnicka B, Chorostowska-Wynimko J. Biomed Pharmacother. 2024 Jun;175:116618.

Reviewer 2 Report

Comments and Suggestions for Authors

The manuscript cancers-3069099 reviews serpinB3. The authors updated this protein with some diseases, e.g., fibrosis and carcinogenesis. The manuscript is well written, and I recommend the publication after minor revisions, as follows:

1) What is the role of mutations in serpinB3? What influences its biological action (more specifically in fibrosis and carcinogenesis)? Please, explore it in the review, e.g., provide a Table summarizing the main mutations, causes, and consequences.

2) In the review, explore the structural motif of serpinB3 and if possible, compare it with serpinB4 (the authors offered a brief description of it). Try to find a correlation structure-activity. For example, the authors can explore the Protein Data Bank containing the serpin structure and superpose some structures with PyMOL software, highlighting the beta-sheets A, B, and C and RSL motifs.

3) In section 4, please provide the chemical structure of the potential inhibitors highlighted by the authors.

Round 2

Reviewer 1 Report

Comments and Suggestions for Authors

The authors responded well to the reviewers' comments, and it is believed that the current manuscript is ready for publication.